

**Technical Note: Improved synthetic routes to *cis*- and *trans*-(2-Methyloxirane-**
**2,3-diyl)dimethanol (*cis*- and *trans*-β-isoprene epoxydiol)**
Molly Frauenheim[1,] Jason D. Surratt[1,2], Zhenfa Zhang[1], Avram Gold[1]
[1]Department of Environmental Sciences and Engineering, Gillings School of Global Public Health,
The University of North Carolina at Chapel Hill, NC, 27599-7431, USA
[2]Department of Chemistry, College of Arts and Sciences, The University of North Carolina at Chapel Hill,
Chapel Hill, NC, 27599-3290, USA
*Correspondence to*: Avram Gold (golda@email.unc.edu)
**Abstract.** We report improved synthetic routes to the isomeric isoprene-derived β-epoxydiols (β-
IEPOX) in high yield (57-69%) from inexpensive, readily available starting compounds. The
syntheses do not require the protection/deprotection steps or time-consuming purification of
intermediates and can readily be scaled up to yield the target IEPOX isomers in gram quantities.
Emissions of isoprene (2-methyl-1,3-butadiene, $C_5H_8$), primarily from deciduous vegetation,
constitute the largest source of nonmethane atmospheric hydrocarbons. In the gas phase under low-
nitric oxide (NO) conditions, addition of atmospheric hydroxyl radical (OH) followed by rapid
addition of $O_2$ yields isoprene-derived hydroxyperoxyl radicals. The major sink (>90%) for the
peroxyl radicals is sequential reaction with hydroperoxyl radical ($HO_2$), OH and $O_2$, which is then
followed by the elimination of OH to yield a ~2:1 mixture of (2-methyloxirane-*cis*/*trans*-2,3-
dilyl)dimethanol (*cis*/*trans*-β-IEPOX). The IEPOX isomers account for about 80% of the closed-
shell hydroxyperoxyl products, and are rapidly taken up into acidic aerosols to form secondary
organic aerosol (SOA). IEPOX-derived SOA makes a significant mass contribution to fine
particulate matter ($PM_{2.5}$), which is known to be a major factor in climate forcing as well as
adversely affects respiratory and cardiovascular systems of exposed populations. Prediction of
ambient $PM_{2.5}$ composition and distribution, both in regional- and global-scale atmospheric
chemistry models, crucially depends on the accuracy of identification and quantitation of uptake
product formation. Accessibility of authentic *cis*- and *trans*-β-IEPOX in high purity and in large
quantity for laboratory studies underpins progress in developing models as well as identification
and quantitation of $PM_{2.5}$ components.




## 1. Introduction

We report here straightforward procedures for the synthesis of isomeric isoprene β-epoxydiols (β-
IEPOX) in high yield from inexpensive, readily available starting compounds. The syntheses do
not require the protection/deprotection steps or time-consuming purification of intermediates as
used in past studies (Cole-Filipiak et al., 2010; Zhang et al., 2012; Bates et al., 2014; Chase et al.,
2015; Bates et al., 2016), and can readily be scaled up to yield the target IEPOX isomers in gram
quantities.

Yearly global emissions of isoprene (2-methyl-1,3-butadiene, $C_5H_8$), primarily from deciduous
vegetation, are estimated to be between 500 and 600 Tg, and constitute the largest source of
nonmethane atmospheric hydrocarbons (Kanakidou et al., 2005; Guenther et al., 2006; Hallquist
et al., 2009; St. Clair et al., 2016). In the gas phase under low-nitric oxide (NO) conditions,
atmospheric hydroxyl radical (OH) adds rapidly to isoprene almost exclusively at C1 and C4,
followed by addition of $O_2$ to yield β- or δ-hydroxyperoxyl radicals (Hallquist et al., 2009). The
major sink (>90%) for the peroxyl radicals is reaction with hydroperoxy radical ($HO_2$) to give
closed-shell isoprene hydroxyhydroperoxides (ISOPOOHs). ISOPOOHs then undergo sequential
addition with OH and $O_2$, followed by the elimination of OH to yield a ~2:1 mixture of (2-
methyloxirane-*cis*/*trans*-2,3-dilyl)dimethanol (*cis*/*trans*-β-IEPOX). The IEPOX isomers account
for about 80% of the closed-shell hydroxyperoxyl products (St. Clair et al., 2016; Wennberg et al.,
2018; Paulot et al., 2009), and are rapidly taken up onto acidic aerosols (Lin et al., 2012; Gaston
et al., 2014; Riedel et al., 2015). IEPOX isomers thus make a significant mass contribution to
secondary organic aerosol (SOA) (Surratt et al., 2010; Riva et al., 2019), and the resulting
atmospheric fine particulate matter ($PM_{2.5}$) (Lin et al., 2013; Budisulistiorini et al., 2015;
Budisulistiorini et al. 2016; Rattanavaraha et al. 2016). $PM_{2.5}$ is known to be a major factor in
climate forcing (Hallquist et al., 2009), and adversely affects respiratory and cardiovascular
systems of exposed populations (Pope and Dockery, 2006; Pye et al., 2021). Advancing the
understanding of the impacts of $PM_{2.5}$ requires the ability to predict $PM_{2.5}$ composition and
distribution, both in regional- and global-scale atmospheric chemistry and climate models (Pye et
al., 2013; McNeill et al., 2015; Marais et al., 2016; Jo et al., 2019), which in turn depends crucially
on the accuracy of identification and quantitation of uptake product formation. As major



precursors of PM$_{2.5}$, *cis-* and *trans-*β-IEPOX have been the focus of considerable effort to elucidate
mechanisms underlying PM$_{2.5}$ formation and aging (Lin et al., 2012; Lin et al., 2014; Gaston et al.,
2014; Nguyen et al., 2014; Zhang et al., 2018; Riva et al., 2019; Armstrong et al., 2022; Cooke et
al., 2022). Underpinning such efforts is the availability of authentic *cis-/trans-*β-IEPOX in high
purity and in quantity.

**2. Experimental**

The esterification and DIBAL-H reduction of mesaconic and citraconic acids generally followed
the procedure reported by Klimovica, et al. (2011). Epoxidation of the *E-* and *Z*-2-methylbut-2-
ene-1,4-diols followed the procedure reported by Zhang et al. (2012). [1]H NMR spectra of isolated
products are provided in the Supplement.

**2.1. *Trans-*β-IEPOX (*trans-*(2-methyloxirane-2,3-dilyl)dimethanol) from mesaconic acid**
*2.1.1. Mesaconic acid, dimethyl ester.* To a solution of mesaconic acid (11.00 g, 84.5 mmol) in
methanol (100 mL), conc. H$_2$SO$_4$ (3 mL) was added. The reaction mixture was refluxed for 8 h
until a complete conversion was observed by thin layer chromatography (TLC) and the reaction
was the neutralized by addition of triethylamine (1.5 mL). Silica gel (11 g) was added and the
resulting mixture was dried on a rotary evaporator under house vacuum. The residue was applied
to a silica gel column and eluted with EtOAc/petroleum ether (1:1) to afford mesaconic acid
dimethyl ester as a colorless oil (13.40 g, 96.5%), purity >98% by NMR. NMR (400 MHz,
chloroform-*d*): δ 6.71 (d, *J* = 1.6 Hz, 1H), 3.74 (s, 3H), 3.71 (s, 3H), 2.23 (d, *J* = 1.6 Hz, 3H),
Figure S1.

*2.1.2. E-2-Methyl but-2-ene-1,4-diol.* A solution of mesaconic acid dimethyl ester (13.40 g, 84.72
mmol) in methylene chloride (120 mL) under argon was cooled to 0 °C and diisobutylaluminum
hydride (DIBAL-H; 400 mL 1.0 M solution in toluene, 400 mmol) was added dropwise, and the
reaction mixture was stirred at 0°C for 1 h. The reaction was diluted with ether (100 mL) and
quenched with water 16 mL (0.04 volume-equivalents of DIBAL-H), followed by 16 mL 15%
sodium hydroxide solution (0.04 volume-equivalents), and 40 mL water (0.1 volume-equivalents).
After quenching, the mixture was allowed to warm to room temperature over 2 h and dried over



magnesium sulfate. The aluminum salt was filtered out through a pad of Celite, and solid filtrant
further washed with ethyl acetate (100 mL). The solvent was removed from filtrate on a rotary
evaporator under house vacuum to yield *E*-2-methyl but-2-ene-1,4-diol as a colorless oil (6.12g,
70.7%), purity > 98% by NMR. NMR (400 MHz, $D_2O$): δ 5.54 – 5.43 (m, 1H), 4.06 (d, *J* = 7.0
Hz, 2H), 3.90 (s, 2H), 1.57 (s, 3H), Figure S2.

*2.1.3. Trans-(2-Methyloxirane-2,3-diyl)dimethanol (trans-β-IEPOX).* Epoxidation of *E*-2-methyl
but-2-ene-1,4-diol followed a published procedure (Zhang et al., 2012). The butene diol (6.00 g,
58.8 mmol) was dissolved in acetonitrile (80 mL) and cooled in an ice-water bath. *m*-
Chloroperoxybenzoic acid (15.53 g, 90 mmol) was added and the clear solution was stirred in the
ice-water bath for 2 h, and then at room temperature until complete transformation of the starting
material as monitored by TLC. The reaction mixture was cooled at 4°C and the resulting precipitate
separated by filtration to remove the bulk of the 3-chlorobenzoic acid. The filtrate was
concentrated under reduced pressure and the residue dissolved in water (30 mL), and washed
repeatedly with chloroform. The aqueous solution was lyophilized to yield *trans*-(2-
methyloxirane-2,3-diyl)dimethanol as colorless oil (6.24 g, 89%), purity > 98% by NMR. The $^1$H
NMR spectrum was identical to that reported in previous syntheses (Zhang et al. 2012). NMR (400
MHz, $D_2O$): δ 3.78 (dd, *J* = 12.49, 4.29 Hz, 1H), 3.59 (d, *J* = 12.56 Hz, 1H), 3.58 (dd, *J* = 12.49,
7.08 Hz, 1H), 3.44 (d, J = 12.56 Hz, 1H), 3.15 (dd, *J* = 7.08, 4.29 1H), 1.23 (s, 3H), Figure S3.
Overall yield for the synthesis of *trans*-β-IEPOX from mesaconic acid was 62.5%.

## 2.2. *cis*-β-IEPOX (*cis*-(2-methyloxirane-2,3-diyl)dimethanol) from citraconic acid

*2.2.1. Citraconic acid, dimethyl ester.* To a solution of citraconic acid (1.98 g, 15.2 mmol) in
methanol (50 mL) conc. $H_2SO_4$ (0.8 mL) was added. The reaction mixture was refluxed for 8 h
until complete conversion as determined by TLC and then neutralized by addition of triethylamine
(0.5 mL). Silica gel (3 g) was added and the resulting mixture was concentrated *in vacuo.* The dry
residue was applied to a silica gel column and eluted with EtOAc/petroleum ether (1:1) to afford
the desired citraconic acid dimethyl ester (2.21g, 92.1%) as a colorless oil, purity > 98% by NMR.
NMR (400 MHz, $D_2O$): δ 6.08 (d, *J* = 1.6 Hz, 1H), 3.85 (s, 3H), 3.75 (s, 3H), 2.07 (d, *J* = 1.6 Hz,
3H), Figure S4.



*2.2.2. Z-2-methyl but-2-ene-1,4-diol.* A solution of citraconic acid dimethyl ester (2.21g, 14.0
mmol) in methylene chloride (25mL) under argon was cooled to 0°C and DIBAL-H (70 mL 1.0
M solution in toluene, 70.0 mmol) was added dropwise. Reaction mixture was stirred at 0°C for 1
h, diluted with ether (60 mL) and quenched with 2.8 mL water (0.04 of DIBAL-H volume
equivalents), followed by 2.8 mL 15% sodium hydroxide solution (0.04 volume equivalents) and
7.0 mL water (0.1 volume equivalents). After quenching, the mixture was allowed to warm to
room temperature over 2 h and dried over magnesium sulfate. The aluminum salt was removed by
filtration through a pad of Celite, and solid filtrant further washed with ethyl acetate (100 mL).
The solvent was removed from filtrate and desired *Z*-2-methyl but-2-ene-1,4-diol (1.07 g, 74.9%
yield) was obtained as a colorless oil, purity > 98% by NMR (Klimovica, et al., 2011). NMR (400
MHz, $D_2O$): δ 5.57 (t, *J* = 7.2 Hz, 1H), 4.16 (d, *J* = 7.2 Hz, 2H), 4.14 (s, 2H), 1.82 (s, 3H), Figure
S5.

*2.2.3. cis-(2-Methyloxirane-2,3-diyl)dimethanol (cis-β-IEPOX).* The epoxidation of the butene
diol was performed according to a published method (Zhang et al., 2012). The butene diol (1.07
g, 10.4 mmol) was dissolved in acetonitrile (25 mL) and cooled in an ice-water bath. *m*-
Chloroperoxybenzoic acid (2.69 g, 15.6 mmol) was added and the clear solution was stirred in the
ice-water bath for 1h, and then at room temperature until complete transformation of the starting
material as monitored by TLC. The mixture was cooled at 0°C and the resulting precipitate
separated by filtration to remove the bulk of the 3-chlorobenzoic acid. The filtrate was
concentrated under reduced pressure, the residue dissolved in water (15 mL) and washed
repeatedly with chloroform. The aqueous solution was lyophilized to give *Z*-(2-methyloxirane-
2,3-dilyl)dimethanol as colorless oil isolated as the crude product (1.03 g, 82.9%), purity 88% by
NMR. NMR (400 MHz, $D_2O$): δ 3.78 (dd, *J* = 12.5, 3.9 Hz, 1H), 3.61 (d, *J* = 12.3 Hz, 1H),3.54
(dd, *J* = 12.6, 7.35 Hz, 1H), 3.52 (d, *J* = 12.3 Hz, 1H), 3.10 (dd, *J* = 7.35, 3.93 Hz, 1H), 1.43 (s,
3H), Figure S6.
Overall yield for the synthesis of *cis*-β-IEPOX from citraconic acid was 57.1%.

**2.3. cis-β-IEPOX (cis-(2-methyloxirane-2,3-diyl)dimethanol) from 3-methyl-2(5H)-furanone**
*2.3.1. Z-2-methyl but-2-ene-1,4-diol.* A solution of 3-methyl-2(5H)-furanone (1.06 g, 10.8 mmol)
in methylene chloride (30 mL) under argon was cooled to 0°C, and DIBAL-H (21 mL 1.0 M



solution in toluene, 21.0 mmol) was added dropwise. The reaction mixture was stirred at 0°C for
1 h, diluted with ether (60 mL) and quenched with 0.85 mL water (0.04 of DIBAL-H volume-
equivalents), followed by 0.85 mL 15% sodium hydroxide solution (0.04 DIBAL-H volume-
equivalents), and 2.1 mL water (0.1 volume-equivalents). After quenching, the mixture was
allowed to warm to room temperature over 2 h, and dried over magnesium sulfate. The aluminum
salt was filtered out through a pad of Celite, and solid filtrant further washed with ethyl acetate
(100 mL). The solvent was removed on a rotary evaporator under house vacuum to yield *Z*-2-
methylbut-2-ene-1,4-diol (1.03 g, 94.1% yield) as a colorless oil, purity > 98% by NMR, Figure
S5.

*2.3.2. cis-(2-Methyloxirane-2,3-diyl)dimethanol (cis-β-IEPOX).* Butene diol (0.9169 g, 8.99
mmol) was dissolved in acetonitrile (50 mL) and cooled in an ice-water bath. *m*-
Chloroperoxybenzoic acid (1.92 g,13.5 mmol) was added and the clear solution was stirred in the
ice-water bath for 1 h, and then at room temperature until complete transformation of the starting
material as monitored by TLC. The mixture was cooled at 0°C and the resulting precipitate
separated by filtration to remove the bulk of the 3-chlorobenzoic acid. The filtrate was dried on a
rotary evaporator under house vacuum and the residue dissolved in water (30 mL). The aqueous
solution was washed repeatedly with chloroform and lyophilized and isolated as the crude product
to give *cis*-(2-methyloxirane-2,3-diyl)dimethanol as colorless oil (0.88 g, 82.9%), purity 88% by
NMR, Figure S6.
Overall yield for the synthesis *cis*-β-IEPOX from 3-methyl-2(5H)-furanone was 69.2%.

**3. Results and discussion**

Several synthetic routes to the β-IEPOX isomers have been published to date. Procedures for the
synthesis of *trans*-β-IEPOX followed the three strategies given in Figures 1 – 3.



**Figure 1.** Synthesis of a *cis*-and *trans*-β-IEPOX mixture from isoprene.

Figure 1, the first published route to *trans*-β-IEPOX (Cole-Filipiak et al., 2010), yielded a mixture
of *cis* and *trans* products in four steps. The procedure is lengthy and required a double vacuum
distillation for isolation of 1,4-dibromoisoprene and a further vacuum distillation for isolation of
the 1,4-diol. The mixture of *cis*- and *trans* epoxides was not separated, and the combined overall
yield was 11%.


**Figure 2.** Synthesis of *trans*-β-IEPOX from prenol.

The approach in Figure 2 (Zhang et al., 2012) has been used in most syntheses reported
subsequent to its publication in 2012. Figure 2 targets synthesis of the *trans* isomer starting with
prenol (3-methyl-2-buten-1-ol). $SeO_2$ oxidation of the trisubstituted olefin yielded *E*-2-methylbut-
2-ene-1,4-diol. Deprotection of the diol, followed by epoxidation with *m*CPBA gave the target
*trans*-β-IEPOX in an overall yield of 43%. The expected *trans* geometry of the ultimate IEPOX
isomer (Trachtenberg, et al., 1970; Sharpless and Lauer, 1972) was confirmed by the absence of a
nuclear Overhauser effect correlation between the methyl group and the oxirane proton in the 1D
NOESY spectrum. An overall yield of ~11% can be calculated for synthesis by the scheme in
Figure 2 in the only other report citing yields (Bates et al., 2014). The $SeO_2$ oxidation/$NaBH_4$
reduction sequence to generate 2-methylbut-2-ene-1,4-diol appears largely responsible for the
discrepancy in yields. Isolation of the diol from the $NaBH_4$ reduction step yields a mixture from
which isolation of product is challenging and is most likely the source of the difference.
More recently, a route to *trans*-β-IEPOX-$d_2$ illustrated in Figure 3 has been reported that could
also serve as a route to the protio compound by substituting of $LAH_4$ for $LAD_4$ as reducing agent
(Chase et al., 2015).








**Figure 3.** Synthesis of *trans*-β-IEPOX-$d_2$, adaptable to synthesis of protio analog.

This route also involves a problematic metal hydride reduction step and an overall yield of 31% was reported. Reaction schemes in Figures 1 – 3 have in common steps that are difficult to accomplish, such as vacuum distillations, or require carefully controlled conditions for protection/deprotection of labile substituents, with the best reported yield being 43% for the reaction Scheme in Figure 2 (Zhang et al. 2012).

Here we report a procedure for the synthesis of pure racemic *trans*-β-IEPOX that is efficient, simple and provides the target IEPOX in an overall yield of 62.5%. No protection/deprotection reactions, which add steps and can decrease yields are involved, and no specialized glassware or instrumentation is required. The strategy for synthesis follows the route in Figure 4, which is based on inexpensive, readily available mesaconic acid as starting material.

222

**Figure 4.** Newly developed route to trans-β-IEPOX starting with mesaconic acid.

224

Mesaconic acid can be esterified to the dimethyl ester in high yield by refluxing in methanol containing 2% concentrated sulfuric acid. The diester is reduced by diisobutylaluminum hydride (DIBAL-H) in methylene chloride to *E*-2-methylbut-2-en-1,4-diol, which is epoxidized by *m*CPBA in acetonitrile. Key to the procedure is the efficient extraction of *E*-2-methylbut-2-ene-1,4-diol from the DIBAL-H reduction reaction with ethyl acetate, which allows recovery of the diol in 70% yield. The synthetic route in Figure 4 will make *trans*-β-IEPOX readily available to laboratories without sophisticated synthesis capabilities. The procedure is particularly attractive because it can readily be scaled up to produce *trans*-β-IEPOX in gram quantities.




The isolation of *E*-2-methylbut-2-en-1,4-diol from the DIBAL-H reduction reaction in high
yield by extraction with ethyl acetate led us to revisit published syntheses of *cis*-β-IEPOX in which
metal hydrides were used as reduction reagents (Bates et al., 2014; Bates et al., 2016; Zhang et al.,
2012). In Figures 5 and 6, citraconic anhydride was the starting material and the reducing agent
was DIBAL-H.

**Figure 5.** Synthesis of cis-β-IEPOX from citraconic anhydride.


**Figure 6.** Streamlined route to cis-β-IEPOX from citraconic anhydride.

The reported overall yield of *cis*-β-IEPOX from synthetic route in Figure 5 was 12% (Bates et al.,
2014). The reaction scheme in Figure 6 streamlined the synthesis through bypassing steps 2 and 3
of the reaction scheme in Figure 5 by direct reduction of citraconic anhydride to *Z*-2-methylbut-2-
en-1,4-diol (Bates et al., 2016). Reduction of the anhydride required forcing conditions (5
equivalents of DIBAL-H were used to achieve reduction of citraconic anhydride) and possibly less
efficient recovery of the diol resulted in the same overall yield reported for the synthesis in Figure

251    5.


3-Methylfuran-2(5H)-one was the starting material for the synthesis in Figure 7 and was reduced
to directly *Z*-2-methylbut-2-en-1,4-diol by LAH (Zhang et al., 2012).

**Figure 7.** Synthesis of cis-β-IEPOX from 3-methylfuran-2(5H)-one.



LAH is a powerful reducing agent leading to some unavoidable over-reduction of the furanone to
the saturated diol, and the overall yield was 19%. We repeated the synthesis of *cis*-β-IEPOX using
either citraconic acid or 3-methylfuran-2(5H)-one as starting points (Figure 8). Because reduction
of anhydrides to diols generally appears to be more difficult and require forcing conditions
(Bloomfield and Lee, 1967), we selected citraconic acid rather than the anhydride as the starting
point for the synthesis. Citraconic acid requires esterification prior to reduction. Although the
dimethyl ester is commercially available, the esterification reaction is very straightforward with a
nearly quantitative yield and the savings in cost is substantial.



**Figure 8.** Synthesis of cis-β-IEPOX from citraconic acid.
Under the assumption that reduction of the *cis* diester would nevertheless not proceed as readily
as reduction of the *trans*-diester, five equivalents of DIBAL-H were used for the reduction of
citraconic diester. With a yield of 74.9%, the reduction of the diester proved to be much more
efficient than that of citraconic anhydride, for which the reported yield was 28% (Bates et al.,
2016). The overall yield for this route was 57.1%, which represents a significant improvement
over the yields reported from the schemes in Figures 5 and 6..

3-Methylfuran-2(5H)-one is a more expensive starting compound than citraconic acid or citraconic
anhydride, but the procedure is streamlined to two steps and the overall yield, at 69%, is much
higher than for any of the published routes. Cost of reagents would largely dictate the choice of
citraconic acid or 3-methylfuran-2(5H)-one starting material. We note that *trans*- and *cis*-β-IEPOX
were isolated directly following lyophilization. The purity of *trans*-β-IEPOX was > 98% by NMR
(Figure S3). *Cis*-β-IEPOX invariably contained a small amount of hydrolysis product (Figure S6)
which may be removed completely by column chromatography if required.



*Supplement:* The supplement related to this article is available online. Included in the supplement are [1]H NMR spectra of the target *cis-* and *trans-*β-IEPOX isomers, as well as the key intermediates in the synthetic routes.

*Author contributions:* MF, AG and ZZ contributed equally to planning and performing experiments and preparation of the manuscript. MF acquired spectroscopic data. J.D.S. helped with editing and manuscript preparation.

*Competing interests:* The authors declare that they have no conflict of interest.

*Financial support:* This project was supported by NSF grant AGS-2001027 (AG, ZZ). JDS and ZZ were supported in part by NSF grant AGS-2039788.

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
