# Peer review of "Technical Note: Improved synthetic routes to *cis*- and *trans*-(2-Methyloxirane- 2,3-diyl)dimethanol (*cis*- and *trans*-β-isoprene epoxydiol)"

_EGUsphere, 2023_

## Author Response (AR1)

Molly Frauenheim
Response to Reviewer Comments
5/13/23

**Reviewer #1**

**General comments**

This manuscript describes the synthesis and characterization of the cis- and trans- isomers of beta-isoprene-epoxydiol.   Specifically, the work refines previously identified routes to these very important atmospheric species.  The procedures are generally well-described, and the compounds are appropriately characterized by NMR.  Because of the need for these starting materials in laboratory atmospheric chemistry work on isoprene-related secondary organic aerosol formation, this work will be of interest to readers of *EGUSphere*.  I recommend publication after the authors consider the following suggestions for improvement of the manuscript.

**Specific comments**

Line 36:  The new procedure does require the purification of an intermediate (the ester), so this statement is not quite correct.  In general, it seems to me that the main advantage of the new routes is higher yield and rather than dramatically reduced effort required.

The procedure of using column chromatography to purify the ester intermediates was adopted from Klimovica et al. (2011). However, because the reaction was typically complete and clean, it was often not necessary to obtain pure product. Thus, we have removed these statements (**Lines 81-82 and 119-120 in preprint**).

Line 83:  I assume that the solvent was removed by rotovap – this should be stated explicitly.

The relevant portion of the text was removed, per the previous comment (**Lines 81-82 in preprint**).

Line 89:  Assuming 1 drop/sec at 0.05 ml per drop, I estimate that this 400 ml addition process would take over 2 hours – is that the correct time frame?

Yes, this is the correct time frame for the reported reaction scale. We have specified this in the revised text, **Lines 84-85.**

Line 94:  Is this a vacuum or a gravity filtration process?

We have specified that the filtration is under vacuum in **Line 89** of the revised text.

Line 104:  About how long did this process take?

Complete transformation of *E*-2-methylbut-2-ene-1,4-diol to *trans-(2-methyloxirane-2,3-diyl)dimethanol* took 3 hours total, comprised of 2 hours at 4°C and 1 additional hour at room temperature, which is reflected in the revised text, **line 98,**

Molly Frauenheim
Response to Reviewer Comments
5/13/23

Line 121:  I assume that the solvent was removed by rotovap – this should be stated explicitly.

The relevant portion of the text was removed, per a previous comment (**Lines 119-120 in preprint**).

Line 267:  What is the rationale for this assumption?

Since DIBAL-H is sterically demanding and the carbonyl groups of the *cis*-diester are hindered, excess DIBAL-H (5 molar equivalents) was used to ensure complete reduction. We have revised the text to reflect this rationale in **Line 255.**

Molly Frauenheim
Response to Reviewer Comments
5/13/23

**Reviewer #2**

This technical note describes an improved synthetic pathway to cis and trans isomers of isoprene-derived β-epoxydiols. These compounds are important precursors to isoprene-derived SOA. The synthetic routes presented provide access to the targeted compounds from available starting materials, in fewer steps, and with higher yields that prior studies. Although purely synthetic organic chemistry topics are not common in this journal, I believe that this topic is highly relevant to our field. I encourage publication after addressing the following concerns:

1. It is surprising that the authors would use only one characterization method for their intermediates and final products (1H NMR). It would be helpful for the authors to include two additional characterization methods, such as 13C NMR and high-resolution mass spectrometry, which would further validate the formation of the targeted molecules and support others in following this synthetic pathway.

We have included $^{13}C$ spectra as well as (-)ESI-HR-QTOF-MS spectra in the Supplement for target *cis-/trans*- β-IEPOX products to further validate the synthetic routes (**Figures S4,S5, S9,S10**).

2. The Results section of the paper includes substantial background information that should be moved to the introduction. Specifically, lines 180-206 and Figures 1-3 describe prior synthetic routes to the trans isomer. Lines 234-258 and Figures 5-7 describe prior synthetic routes to the cis isomer.

We adopted the current organization because the material included in the lines cited by the reviewer explains the advantages of each route over previous synthetic reports of each of the compounds. Reorganization as suggested by Reviewer 2 would fragment this narrative. The introduction is intended to emphasize the importance of thesynthetic targets to the atmospheric chemistry community. For this reason, we prefer to keep the present organization.

3. Can the authors add a paragraph on the safety of these methods, toxicological information of the products, and discussion of any hazards specific to this process and how they can be mitigated. I suspect that these reaction pathways may be of interest to atmospheric chemists who may have limited experience in organic synthesis, given that they are described as being relatively accessible. Or if these methods should only be conducted by experienced synthetic organic chemists, please state that.

Chemical and toxicological hazards of starting materials are publicly available via the chemical safety data sheet (SDS, https://chemicalsafety.com/sds-search/) and there are no exceptionally toxic or hazardous materials. Neither *m*CPBA nor DIBAL-H are explosion hazards unless heated. If carried out under the conditions (temperature, inert atmosphere, reaction time, etc.) as outlined in this manuscript, the reactions are safe and straightforward. However, this information is not known for intermediates and products, as they are not commercially available.

Synthesis reactions should be carried out by a student who has taken an undergraduate course in introductory organic chemistry. General safety protocols and chemical handling learned in this course will be sufficient to conduct the reactions safely.

Molly Frauenheim
Response to Reviewer Comments
5/13/23

We have added the statement (**Line 64 revised copy**): All reactions should be performed under a fume hood.

The quality of Figure S1 in the Supporting information should be improved, because it appears blurry. Peak integration is missing.- add this

Figure quality has been corrected in S1 and peak integrations have been added to all [1]H NMRs in the SI (**Figures S1-4, S6-8**).

4. Line 79, 105, elsewhere - Please describe the conditions of TLC (i.e. plate and solvent).

TLC conditions have been added to revised text, **Lines 77,99, 113, 135,159.**

5. Line 19, some words appear to be missing in this description.

No words are missing in the description.

6. Line 121, 133, how was the solvent removed? Please indicate that equipment and conditions (ambient versus vacuum) were utilized.

We have specified that the filtration is under house vacuum in **Lines 89, 101** of revised text.

7. Cost savings of this approach is mentioned as being an improvement in relation to prior synthetic routes. Can the authors provide some quantitative assessment to support this claim?

We suggest that the cost of starting materials is generally inexpensive. To support the claim that costs of starting reagents are relatively inexpensive, we have included the respective costs at the time of publication in **Lines 75, 112.**

There will be associated cost savings for the proposed routes compared to previous routes due to the reduced number of steps involved. Constantly changing costs and availability of starting materials and reagents does not justify a more detailed analysis.

Molly Frauenheim
Response to Reviewer Comments
5/13/23

**Magda Claeys**

This is a fine study reporting convenient synthetic routes to cis- and *trans*-β-isoprene epoxydiol, which are useful for diverse laboratory studies. The organic synthetic routes are well thought off, make use of readily available starting materials, require no laborious protection/deprotection steps and extensive purification steps of intermediates, and lead to gram amounts of the targeted epoxydiols. I only have some very minor technical comments.

**Technical comments:**

1. Title: chemical names are usually not capitalized. I would write: *"....cis- and trans-*(2-methyloxirane-2,3-diyl)dimethanol …"

The suggested title format has been applied.

2. Number of significant digits: I suggest to reduce the number of significant digits to 2 or in case a number starts by "1" to 3. For example, line 77: "… mesaconic acid (11.0 g, 85 mmol) …."

The number of significant digits has been reduced to 2 as suggested throughout experimental section of the main text.

3. Figures: it would be useful to also explain the abbreviations used in the schemes. For example, Figure 1: the abbreviation *m*CPBA is explained in the text but it would be useful to repeat this in the legend to the figure.

Because *m*CPBA is defined in Line 105 on first use and is a commonly used reagent abbreviation, we do not feel the need to further define the acronym in the figure captions.

4. References: titles of journal articles are usually not capitalized.

References have been edited such that journal titles are not capitalized.